# Exploring Machine Learning-Based Fault Monitoring for Polymer-Based Additive Manufacturing: Challenges and Opportunities

**DOI:** 10.3390/s22239446

**Published:** 2022-12-02

**Authors:** Gabriel Avelino R. Sampedro, Syifa Maliah Rachmawati, Dong-Seong Kim, Jae-Min Lee

**Affiliations:** 1Department of IT Convergence Engineering, Kumoh National Institute of Technology, Gumi 39177, Republic of Korea; 2College of Computer Studies, De La Salle University, 2401 Taft Ave., Malate, Manila 1004, Philippines; 3Faculty of Information and Communication Studies, University of the Philippines Open University, Los Baños 4031, Philippines

**Keywords:** 3D printing, nozzle clogging, machine learning, smart monitoring

## Abstract

Three-dimensional printing, often known as additive manufacturing (AM), is a groundbreaking technique that enables rapid prototyping. Monitoring AM delivers benefits, as monitoring print quality can prevent waste and excess material costs. Machine learning is often applied to automating fault detection processes, especially in AM. This paper explores recent research on machine learning-based mechanical fault monitoring systems in fused deposition modeling (FDM). Specifically, various machine learning-based algorithms are applied to measurements extracted from different parts of a 3D printer to diagnose and identify faults. The studies often use mechanical-based fault analysis from data gathered from sensors that measure attitude, acoustic emission, acceleration, and vibration signals. This survey examines what has been achieved and opens up new opportunities for further research in underexplored areas such as SLM-based mechanical fault monitoring.

## 1. Introduction

Additive manufacturing (AM) is a process that fabricates objects by fusing materials in discrete planar layers [1]. The process involves developing a tangible model of a computer-aided design (CAD) file through a program that sends instructions to the printer to perform a layer-by-layer development of the object [2,3]. In turn, the printer that receives the instructions produces a layer-by-layer output to form a 3D object using plastic, metal, glass, or ceramic. AM technologies include powder bed fusion, material extrusion, directed energy deposition, vat polymerization, binder jetting, and sheet lamination [4,5,6]. Each printing process is unique to materials, where melting or hardening is involved in the process. Figure 1 illustrates the different processes that may occur in additive manufacturing.

Due to its ability to recreate tangible objects using raw materials, its applications are endless. Its applications are not limited to industrial settings, as it is employed in areas including biomedical engineering, and civil engineering [7,8]. Additive manufacturing solutions are seen for orthopedic use in recreating body parts, creating machine parts, and even constructing houses. It reduces the time and cost of developing a product before mass production. Furthermore, it enables quick modifications of parts during the testing of a product. AM is efficient in producing small batches, but when manufacturing at high volumes, the production time per unit is too long. Furthermore, layer-based fabrication in AM causes material anisotropy in the produced parts. Thus, the produced parts are not as smooth as those manufactured using factory-grade equipment such as injection molding [6]. Another drawback is the lack of design knowledge and tools for development [9].

While AM is ideal for rapid prototyping and a small-lot alternative to mass manufacturing, the process could be better. The production process in AM can take a few minutes to hours and days. During this lengthy process, a lot can happen to the printing machine and, in turn, affect the output. Machine errors such as overheating, jamming, or mechanical failure must be resolved before producing quality output. In addition, machines are often incapable of detecting faults on their own. The machine will continue to print despite any unfavorable conditions within the machine. Consequentially, the continuous printing process under unfavorable conditions can further damage the machine and output. Thus, it is crucial to monitor the machine for errors constantly. Figure 2 illustrates a common error in FDM printing.

In this paper, the researchers focus on exploring machine learning-based sensor-based monitoring for FDM printers. FDM is a method of AM that forms an object using an extruder that melts and patches together thermoplastic materials. Its output can be crude but reliable enough for simple prototyping [6,10]. Figure 3 illustrates how an FDM printer feeds plastic into the extruder using a set of motors. In the printing process, errors and misprints are often caused by mechanical faults in the machine. In printing, faults may occur when there are issues such as misalignment of the print bed, clogging and misalignment of the nozzle, and the miscalibration of equipment [11]. Given the vulnerabilities to mechanical faults, there is a need for more accurate monitoring of mechanical faults in AM. This is currently a work in progress, as addressed in various studies [1,12,13].

Different methods for identifying mechanical faults have been proposed, and each has its strengths and weaknesses [5]. The most common method for monitoring faults is through manual inspection. The process may be tedious and require time and money, as operators must constantly monitor devices. Unfortunately, manual inspection does not comprehensively analyze the overall machine condition. Sometimes, a fault occurs because of ordinary wear-and-tear, where minor malfunctions progressively cause bigger ones. Heuristics or rules of thumb have traditionally addressed this problem, suggesting that machine performance is influenced by working conditions and allows one to discriminate between normal and faulty situations. Solutions often involve formula- and threshold-based monitoring, but they are not always accurate. Other studies have tried to apply physics-based modeling in analyzing FDM printers. While physics-based modeling is one of the simplest and quickest to implement, this will only work under ideal and controlled operational conditions [14]. In physics-based modeling, readily-available laws of physics and mechanics of materials are applied in analyzing material and component conditions [15]. Ideal conditions are assumed, and only data collected from the sensors connected to the device are considered. While physics-based methods work theoretically, many factors are difficult to consider in reality. Factors such as machine degradation, unmeasured environmental factors, and sensor quality are just some of the many factors contributing to the randomness of the margins of error in analysis [16]. To address such, finite-element modeling (FEM) has been explored in different studies because it can be fully customized to include all necessary aspects and account for process uncertainties [17,18]. In FEM, higher-order differential equations are constructed to estimate the process behavior and the overall geometry of the manufactured parts [19]. The process is intricate (involving computation and optimization of different simultaneous equations) [20,21]. However, it has been proven effective in helping to identify the current and future conditions in the manufacturing process and its ability to respond to design changes. Simpler methods using machine learning have been the focus of recent studies due to their capacity to provide a rapid solution to the modeling processes without the need for mathematical modeling. Instead of coming up with equations, datasets are obtained by running multiple instances of the printing process. The process also takes time and resources, but only for training. In machine learning, historical data is used to model the behavior of machines. Instead of formulating equations or applying physics-based laws, the machine learning model allows systems to develop formulas for computation. Machine learning allows the consideration of certain factors which contribute to the randomness of results. These methods are advantageous for manufacturers because they can be executed faster than FEM and high-end modeling software. For machine learning, the quality of results would still depend on the quality of the training dataset or historical values used to recreate the prediction model. Using historical data, the machine learns from its mistakes and generalizes them by producing a rule [1]. For instance, some proposed solutions used neural network algorithms and device sensors to analyze the various parameters related to the printing operation, such as the attitude signal [22,23], nozzle temperature [24], acoustic emission [25], vibration readings [26], fumes, and even brightness to further monitor the printer for errors [1,5,13,27]. Various algorithms are being examined to correlate different sensor values and predict the presence of a fault within a given system.

In the machine learning-based monitoring system, all sensors will connect to the embedded system unit, such as Raspberry Pi. As the development board, Raspberry Pi will collect all the data from sensors. The sensors will continuously sense the environment. The gathered data will be filtered and preprocessed before being fed into the neural network. The classification result will show in the personal computer whether that timestamp contains fault or not in the data visualization part. The prediction process becomes more complex each time new monitoring data are added, so the model must be retrained. The overview model of the fault monitoring is shown in Figure 4.

This survey provides an overview of current studies and progress in ML solutions for FDM mechanical fault monitoring. In terms of the systems, this paper surveys the following:studies which proposed machine learning-based solutions for fault analysis in data-driven systems;recent studies for monitoring print defects, e.g., delamination, warping, and geometry defects; andrecent studies for diagnosing and predicting faults based on mechanical parts of the 3D printers, e.g., synchronous belts and bearings.

The relevant research papers published between 2017 to 2021 were collected using Mendeley and Google Scholar. The search was performed using keywords such as: “fused deposition modeling,” “additive manufacturing,” and “3D printers”. These terms were linked with combinations including “sensor-based,” “fault monitoring,” “machine learning,” “quality monitoring,” “diagnosis,” and “error.” The retrieved papers were filtered only to include research that involves fault monitoring on FDM printers. The seminal papers about ML and an overview of AM technologies were excluded from the process. Table 1 presents the list of data-driven AM papers categorized by publication years. Most of the research papers were published in 2019 and 2021, followed by 2018.

Table 1 shows that our search only produced two relevant results for 2017 publications. Therefore, it can infer that 2018 is when the research interest in machine learning-based AM monitoring started to pick up. Compared to other publication years, 2019 and 2021 have the highest publication count, with a total of 15 publications each. 2018 closely follows this, with a total of 12 publications. 2020 was observed to decline in the number of published articles significantly. The decline may be attributed to the reduced research productivity due to the COVID-19 pandemic [69]. This survey only considered 2017 to 2021 for the collection period. Based on the data, it is evident that machine learning-based AM monitoring gained traction in 2018, and 2017 is the estimated beginning period of research in the area.

This survey is organized as follows: Section 2 introduces the overview and concepts of AM, which will be the basis for the preceding sections. Section 3 discusses and reviews studies on machine learning-based solutions for mechanical faults. Section 4 tackles the insights and trends observed from the reviewed studies. Finally, Section 5 concludes this survey.

## 2. Equipment

In Table 2, the experimentation equipment used by the reviewed studies was listed. The references listed exclude research with the same machine learning algorithms applied. This table summarizes the printer details, sensor details, and materials used. These details were included to consider the variations in equipment parameters that may be introduced in future experimentation. It also adds to the variables that future research may consider when comparing their methods to past methods. “Printer Details” include the 3D printer maker and the specific model. “Sensor Details” list the sensor brand and models of all the sensors used in the experiments. The “Materials Used” shows the filament or fabrication materials used for the 3D printed models.

There are multiple types of 3D printers based on various processes, such as filament filtration and powder sintering, to stereolithography and powder binding. While there are many technologies, there are also different FDM 3D printers. The four main types are cartesian, delta, polar, and robotic arm. In this study, the 3D printers used in the papers are generally split into SLD-BL600-6 3D delta printer and Hyrel FDM printer. The SLD-BL600-6 3D delta printer is based on delta technology and operates with Cartesian coordinates. This involves a round printing plate combined with an extruder fixed at three triangular points. Each of the three points then moves up and down, determining the print head’s position and direction. Delta printers were designed to speed up the printing process. However, many believe this type of printer is less accurate than a conventional Cartesian printer. The Hyrel FDM printer is a Cartesian printer, the most common type of FDM 3F printer found on the market. Based on the Cartesian coordinate system in mathematics, this technology uses three axes, X, Y, and Z, to determine the correct positions and direction of the print head. With this printer, the printing bed moves only on the Z-axis, with the print head working two-dimensionally on the X–Y plane. The difference between the Cartesian and Delta type of printers is that each element can move in relation to the print bed. In Cartesian 3D printers, each element can only move in one direction, whereas in Delta 3D printers, the printer head can move in any direction, but the print tray does not move.

Attitude signal and acoustic emission (AE) are the most common data signal types used in mechanical fault diagnosis. All the respective papers used the same attitude sensor brand, and monitoring target except for [39]. Studies using the SLD-BL600-6 3D delta printer include [22,23,39,41,44,46]. Studies using the Hyrel FDM printer include [38,60,68], all of which studied AE-based mechanical fault monitoring and used similar experimental setups to analyze extruder states. In 3D delta printers, studies used attitude sensors to acquire the rotation data of each axis.

Due to the number of sensors, studies focused on 3D delta printers generally use low-cost attitude sensors to lessen the cost of the monitoring system. Unlike other mechanical fault monitoring studies, Ref. [39] used an electronic compass to acquire magnetic field data and stated that this sensor could be another option for a low-cost sensor due to its competitive performance.

## 3. Additive Manufacturing and Data-Driven Preparation

This section provides an overview of the printing parameters and equipment, such as materials and sensors in additive manufacturing. Furthermore, it details predictive methods for AM and concludes with a discussion on the common collection procedures and preprocessing for data-driven monitoring.

### 3.1. Sensor, Signal, and Data

Sensors are widely used in modern manufacturing systems to monitor machine health conditions and processes, which produce essential data [71,72]. The data from sensors are often raw and in measurements such as current. For processing sensor data for real-time diagnostics redundancy, numerous parameters have to be analyzed [38]. To develop predictive models for machines, data collection of the mechanical health condition is needed [73,74]. In AM, sensors are attached to measure and detect thermal, acoustic, optical, and ultrasonic signals, which return valuable information [5]. The following are the data used in the monitoring of mechanical faults in 3D printers:Temperature signals yield information about the temperature of mechanical components of machines. Most recent studies using this signal aim to predict delamination and geometry error. Some sensors that can collect this data include thermocouples [58,75], and IR sensors [75].Acoustic emission (AE) signals contain information on mechanisms such as friction, crack, and deformations of produced products. Furthermore, AE sensors can be attached on the side of the filament extruder to collect machine state and determine whether the machine is operating in normal condition or if errors such as clogging and a lack of materials for production [38]. Most studies use this type of sensor to analyze mechanical faults, such as in [38,59,60,70].Attitude signals consist of information about the axes and angles of attitude sensors. Attitude sensors are commonly used in 3D delta printers due to the 3-axes of the printer. Most studies used this type of sensor, as it is low-cost [22,23,41,44,46].Acceleration signals produce information about the extrusion head’s change of motion, acceleration, and vibration. Reference [39] uses acceleration sensors for data-driven monitoring.

### 3.2. Data-Driven Predictive Monitoring

Predictive AM monitoring can be divided into two: physics-driven and data-driven. The physics-based models are used to analyze part geometry, deposition patterns, and power intensities [76,77]. Generally, image processing to this extent would require substantial computational costs to analyze data of that extent. Thus, it is impractical to use physics-driven methods, where data-driven models come into play. On the other hand, data-driven models are based on machine learning models that aim to predict the printer part states [5,78]. Recent advancements in computing and ML pave the way for faster and more accurate algorithms. This allowed the implementation of ML within the field of predictive data-driven AM. Applications of ML within this area may optimize the 3D printing process by helping printer operators predict if the parts of an AM machine will still be in ideal conditions, thereby decreasing errors and predicting faults. In turn, this will decrease the cost of manufacturing and reduce printing errors [1]. Technological progress in recent years provided researchers in this area access to a large amount of data, allowing them to utilize more advanced data-driven paradigms.

## 4. Data-Driven Mechanical Fault Monitoring

This section reviews studies about data-driven monitoring for mechanical fault diagnosis and monitoring. They are grouped based on the type of data and sensors used in the experiments. Hence, this section is divided into four: acoustic emission, acceleration and vibration, attitude signals, and magnetic field. Table 2 lists the summary of the research’s details, equipment, and performances in consideration.

### 4.1. Acoustic Emission

This subsection reviews studies that experiment and analyze the sounds produced by the printing process using AE sensors, including the works of [38,60]. Both studies used similar equipment, analyzed extruder states, and proposed a clustering algorithm for the task. Machine states can be identified based on the AE signal through the extruder that emits materials. The extruder can be identified to be either under normal extruding conditions, semi-blocked, blocked, loading in new materials, or running out of materials to be extruded. Liu et al. used clustering-by-fast-search-and-find-of-density-peaks (CFSFDP), while Lu et al. proposed the K-singular value decomposition (K-SVD) algorithm [38,60]. Before using CFSFDP, Ref. [60] applied dimensionality reduction using linear discriminant analysis (LDA) to the data and identified that reducing feature space dimension to two dimensions has the best performance. Aside from comparing CFSFDP to other unsupervised learning algorithms, the method is also compared to supervised learning algorithms. In comparison, the classification results of CFSFDP are slightly better than some supervised methods. Liu et al. stated that supervised classification methods have better identification than unsupervised methods but come at the cost of a higher computation [60]. Despite being able to identify states, there still exists some difficulty in identifying the semi-blocked state of an extruder.

Lu et al. proposed a physics-constrained dictionary learning approach to improve the efficiency of data processing and perform diagnosis depending on what sensor data produces irregular data [38]. The study focuses on data optimization from AE signals to classify and predict mechanical problems in 3D printers, such as nozzle clogging, excessive vibration, or even over- and under-extrusion. Applying K-SVD to identify a printer’s operational condition is plausible, but its performance differs depending on the variance of data [79,80]. The major challenge of this method comes from the K-SVD algorithm, which can only find the local minima, as not all fault-related issues occur in this area.

Wu et al. proposed a data-driven machine monitoring using a hidden semi-Markov model (HSMM) and AE [70]. The research used the FDM printer for real-time machine state classification. This study extended their previous work, which used a support vector machine (SVM) for classification [81]. HSMM was fed with the data collected from the AE sensors to identify the machine states, which are either normal or abnormal. Furthermore, it produces an identification accuracy rate of 91.9%. When comparing HSMM to the previous method (SVM), HSMM has the advantage of further identifying the cause of the abnormal state condition through multi-state identification. Lastly, it is noted that HSMM provides a more in-depth analysis through machine state classification compared to SVM. Most incorrect identifications by HSMM were due to the overlaps of the data due to the multiple sensor inputs. Further investigation into AE signal noise removal or suppression may be considered for more accurate state identification.

### 4.2. Acceleration and Vibration

In this subsection, the papers are grouped based on their use—accelerometer or vibration sensors. The examined works include that of [26,45] for the vibration sensor and [59] for the accelerometer. While the previous subsection explains the use of AE for mechanical state classification, the works in this section use a different approach to mechanical fault monitoring. However, instead of solely relying on an AE sensor setup, the proposed system adds the acquisition of the acceleration of the printer extruder from an accelerometer sensor attached to it [59]. They used the SVM model to identify if there were any loose bolts within the extruder, which is one of the factors of layer shifting. In this case, the attachment location of the accelerometers is on the nozzle head and build plate. SVM yields either 0 (normal operational states) or 1 (faulty mechanical states). During their faulty state validation experiments, it correctly identifies six out of eight cases. The final accuracy was 87.5% for the real-time experiments conducted on eight healthy and eight faulty states. In this case, their method could save material and energy costs by 75% compared to a non-SVM model.

Li et al. analyzed the build platform and extruder. However, they used a pair of piezoelectric sensors to collect vibration signals for fault monitoring by using least squares SVM (LS-SVM) and back-propagation neural network (BPNN) [45]. LS-SVM identifies filament jamming states, while BPNN determines the presence of abnormalities in the structure of the printed object. They apply synthetic acceleration to reduce the computational complexity and reveal more data; however, there are limitations in considering vibration intensity affected by the feed rate.

Fang et al. propose using Bayesian convolutional neural networks (BCNNs) to classify faulty motor movements [26]. BCNNs are used to determine irregularity in the motor’s vibration sounds, which often indicates an error. An abnormal noise from excessive vibration often signifies motor malfunctions. Usually, noisy motors result from loose parts and thus cause unnecessary movements. Their experiments show that the BCNN has classification and is much more ideal in fault detection based on motor vibration sound compared to the other baseline algorithms. The highest accuracy BCNN yields are 99.82% which is on a non-noisy data set. Their study proves the potential of using uncertainty quantification for misjudgment classification.

Sampedro et al. proposed a real-time fault diagnosis system using a temporal convolutional network (TCN) using the encoder-decoder mechanism of the 3D printer [2]. TCN is a type of convolutional network that handles time-series data. The proposed model consists of an encoder for extracting features from the data and a decoder for recreating the desired output based on the interpretation of the extracted features. The TCN structures for processing multivariate time-series data in detecting mechanical faults. This method placed a set of collaborative sensors on the top of the extruder to monitor vibration signals. When compared against other algorithms such as long short-term memory (LSTM), SVM, and CNN, the TCN model yields a prediction accuracy of 97.2%.

Kim et al. studied data-driven FDM process monitoring and diagnosis based on the SVM model and k-fold cross-validation [59]. In their work, an accelerometer and an AE sensor are used to gather data. The research attributes layer shifting errors to loosened bolts within the machine. When bolts are lost, motors have excessive movements, and the expected print output may not be achieved. In this research, the vibrations that cause loosened bolts are measured through the accelerometer that measures the machine’s x-, y-, and z-coordinates. Two accelerometers are attached to the nozzle head for the x- and y-axes, and another accelerometer is attached to the build plate for the z-axis signals. In addition, AE sensors were attached to the top frame of the 3D printer. The fault monitoring algorithm uses SVM for its effective classification, and root mean square (RMS) to diagnose the condition of the bolts of the printer. SVM is fed with the RMS values of the acceleration and AE data; this yields a numerical output of either 0 (healthy states) or 1 (faulty state). On average, the non-linear SVM model returned a prediction accuracy of at least 77.5%, 82.5 %, and 100% for x-axis acceleration, y-axis acceleration, and AE, respectively. During their faulty state validation experiments, six out of eight cases were correctly identified. Correct identification of faulty cases can save material and energy, as printing faulty outputs may be avoided. The final accuracy was 87.5% for the real-time experiments conducted on eight healthy and eight faulty states. In this case, their method could save costs by 75%.

Scheffel et al. proposed using CNN for online monitoring and fault detection using vibration signals as the input [35]. Their work presents an architecture to store and retrieve vibration readings using an accelerometer from an FDM-type 3D printer. The system makes use of Internet of Things (IoT)-based technology for the real-time collection of data as well as analysis. The synchronized data is used to train a CNN to determine if the patterns measured are irregular and may lead to faulty print output. This study showed good accuracy, recall, and precision rates using CNN to detect vibration patterns, mainly in printing slim parts. The main advantage of this study is that their findings guarantee that the proposed IoT classifier can rely on the data it is processing without forged data. When compared against other algorithms, they achieved an accuracy rate of 97.7% in identifying irregular vibrations.

### 4.3. Attitude Signals

This subsection discusses studies that mainly used attitude signals to the data for data-driven mechanical fault diagnosis. All of the studies here used a 3D delta printer and attitude sensor attached to the printer extruder. The sensors in these studies measure parameters such as 3-axial attitude angle, angular velocity, vibration, and magnetic field intensity for both joint bearing and synchronous belts in 3D printers [56,82].

Zhang et al. proposed error fusion of the multiple sparse auto-encoder (EFMSAE) and deep fuzzy echo state network (DFESN), respectively [22]. In [22], squared prediction error (SPE) is used to evaluate printing degradation performance based on the attitude signals of the printer extruder. This research compares EFMSAE to LSTM; however, LSTM cannot identify the defect promptly. For defects to be determined, the collected data must be preprocessed and transformed into a different form. In [23], DFESN can use the collected data without preprocessing. For the features, they considered 10 patterns for bearings and 16 patterns for 3D printers. These patterns represent various conditions of the extruder concerning the print quality. The results show that DFESN performs better than SAE, ESN, and CNN.

He et al. proposed the application of the echo state networks (ESN) of [83] to diagnose mechanical faults in the 3D printer using low-cost sensors [44]. A 3D delta printer and a low-cost attitude sensor costing 20 USD were used in the tests conducted. The sensors acquired data from the joint bearing and synchronous belt. The proposed method was compared to SVM, PCASVM, and LPPSVM. The results show that ESN has the highest mean accuracy of 97.17%.

Wang et al. used locality preserving projects (LPP), and SVM for monitoring the joint bearing and synchronous belt of the delta 3D printer [41]. The health conditions of the printer are collected using the data acquired by the attitude sensors. LPPSVM, a multi-class SVM classifier using the least-squares loss function, is used to monitor faults. The proposed method was compared to KLPPSVM, NPESVM, and SVM-only models. When utilizing all channel space, at the highest performance, the LPPSVM model achieved an accuracy rate of 93.49%, while the SVM model produced 86.15%.

### 4.4. Magnetic Field

The majority of mechanical fault diagnosis studies use non-magnetic field data. However, in [39], magnetic field signals are analyzed for 3D printing mechanical fault diagnosis. This study proposes using an electronic compass as an alternative low-cost sensor. However, magnetic data makes supervised learning inapplicable due to the expensive cost of obtaining signal acquisition from faulty mechanical systems [84]. One-shot Learning (OSL) based on bidirectional generative adversarial networks (BiGAN) is used to resolve this issue. BiGAN maps the signal input space into a feature space to classify the mechanical health condition.

Furthermore, OSL is compared to both unsupervised and supervised methods. Compared to the unsupervised approach, OSL only requires 60% of the training data to overcome other methods’ sensibility. As seen in the results of this research, the proposed method produces an accuracy of 92.07% ±1.24. For the supervised approach, only one signal in each fault condition is required for classification. It also required less data for mapping. This study proves that magnetic signals could be an alternative to vibration and acoustic emission signals.

## 5. Discussion

This section analyzes and discusses observations from the collection and reviews the papers for AM monitoring. The various equipment used in the experiments and how it relates to further studies are discussed. Finally, it concludes with an analysis of the recent studies, various methods, and trends for AM monitoring research.

### 5.1. Monitoring and Diagnosis

Acceleration, vibration, attitude, and AE signals are the main setups of the studies. Since 3D printers have many moving parts, attachment of attitude sensors may yield high costs for these systems. For mechanical fault detection, some studies opt to use low-cost attitude sensors. These studies proved that low-cost attitude sensors are still viable for predictive mechanical fault diagnosis. For SVM performances, Table 2 shows SVM [59], LPPSVM [41], and TSVM [46], which yields 87.5%, 93.49%, and 84.29%, respectively.

Only one study uses the magnetic field signal, which can be an alternative low-cost sensor solution. Table 2 and Figure 5 compare magnetic field sensor with OSL to attitude sensor with TSVM, LPPSVM, ESN. The algorithms achieved an accuracy rate of 92.07%, 84.29%, 94.49%, and 97.17%, respectively. The supervised ESN-based algorithms such as DFESN [23] and ESN [44] achieved an accuracy rate of more than 90% and 97.17%, respectively. This shows that using a magnetic field signal with OSL has a competitive performance.

Figure 5 shows the performance of four studies that analyze and monitor joint bearing and synchronous belts. The studies analyzed the attitude signals collected using an attitude sensor [41,44,46], except for [39] which analyzed the magnetic field acquired from an electronic compass. Li et al. proposed using magnetic field analysis for mechanical fault monitoring and to assess if it is feasible and has a competitive performance compared to attitude sensors [39].

### 5.2. Challenges and Opportunities

Only a few studies consider latency or the speed of monitoring, such as in [45]. AM monitoring requires accuracy, low-cost sensor solutions, and monitoring speed. This would enhance the reliability of the Internet of Things device through an advanced monitoring system by performing real-time experimentation with the testbed deployed. Some studies applied feature reduction to lighten the computational cost needed to train the algorithm [45]. Real-time AM monitoring would benefit from latency and a lighter computational load. Further research in this field will improve real-time data-driven AM monitoring, which the AM industry needs.

Failures can be caused by several reasons, such as over-extrusion, stringing, and overheating. Besides predicting the normal or abnormal, showing the reason can improve the reliability of the overall monitoring system. Multi-class classification problems can be adopted for further study to differentiate the exact causes of the failures. Regarding this, the deep learning algorithm is an example of a neural network that supports multi-class problems. Better indicators were added, such as precision, recall, and f1-score.

Another challenging concern is the concept of drift when the target variable’s statistical characteristic changes with time. There will be a circumstance when the model in use can no longer make accurate predictions since the meaning of the input data it was trained on has changed over time, while the model in use needs to be made aware of the change. So, concept drift can affect the performance of the prediction accuracy, and it needs a model that can be adaptive to the changes.

Besides building an adaptive model, fixing the distribution data problem can also be the central issue in prediction. The collected instances from sensors may contain a particular class, which is meant for majority class instances. In contrast, the fault data are rare in collected data classified as a minority class. This condition is known as imbalanced data. It can affect performance by misclassifying the data. The existing approaches, such as sampling or synthetic minority over-sampling technique (SMOTE), have been applied to balance the class distribution. However, it is still an open issue in current studies.

The combination of methods in developing better machine learning algorithms is still being carried out. For example, a combination of deep learning and transfer learning, well known as the deep-transfer learning (DTL) model, is a new trend in machine learning. Deep learning has an advantage in feature representation but relies on a large amount of labeled training data. Transfer learning can be added to overcome limitations by emphasizing basic learning information. Therefore, transfer learning can create a model with better performance, as training data is limited.

Furthermore, most research in this area uses vibration, acceleration, or AE signals for their analysis. Looking at Figure 5, it can be inferred that magnetic field signal is a viable method based on the result of [39], with an accuracy rate of 92.7%. Comparing it to others with a similar setup and monitoring type [22,23,41,44,46], the performance in [39] is competitive. Further research comparing the sensors with the same setup would bring new knowledge and opportunity for cost reduction and improvement in mechanical fault diagnosis.

## 6. Conclusions

This survey focuses on machine learning-based predictive monitoring systems for mechanical faults in FDM 3D printers. The researchers collected relevant publications from 2017 to 2021 and observed that 2017 returned minimal articles. 2018, 2019, and 2021 are the primary providers of relevant papers.

It was observed that almost all of the papers garnered conducted FDM-based experiments. These papers are categorized by the type of sensor and signal they use. Specifically, they are divided into four: AE, acceleration and vibration, attitude signals, and magnetic field. Studies that use AE signals proposed using CFSFDP, K-SVD, and HSMM algorithms for mechanical state classification. In unsupervised learning, HSMM and CFSFDP yield an accuracy rate of 91.9% and 79.53–88.0%, respectively.

Studies that use acceleration and vibration signals use SVM, LS-SVM, BPNN, and BCNN for mechanical fault diagnosis. For studies that focused on attitude signals, they used 3D delta printers as the main equipment. These studies proposed EFMSAE, ESN, and DFESN.

In this area, there is a lack of SLM-based mechanical fault studies. Future studies may use their efforts to add a knowledge base here. The mechanics of SLM printers are somewhat similar, as a set of stepper motors is involved in the printing process. Furthermore, vibrations, current intensities (in melting capacity), and attitude signals are also parameters of interest in SLM printing. Furthermore, for the system speed, future studies may also consider including the latency of the systems for real-time monitoring. This topic will be a great study to focus on, as it will be a considerable time and cost-saving if the AM monitoring can change parameters or stop if there are anomalies in the prints.

## Figures and Tables

**Figure 1 sensors-22-09446-f001:**
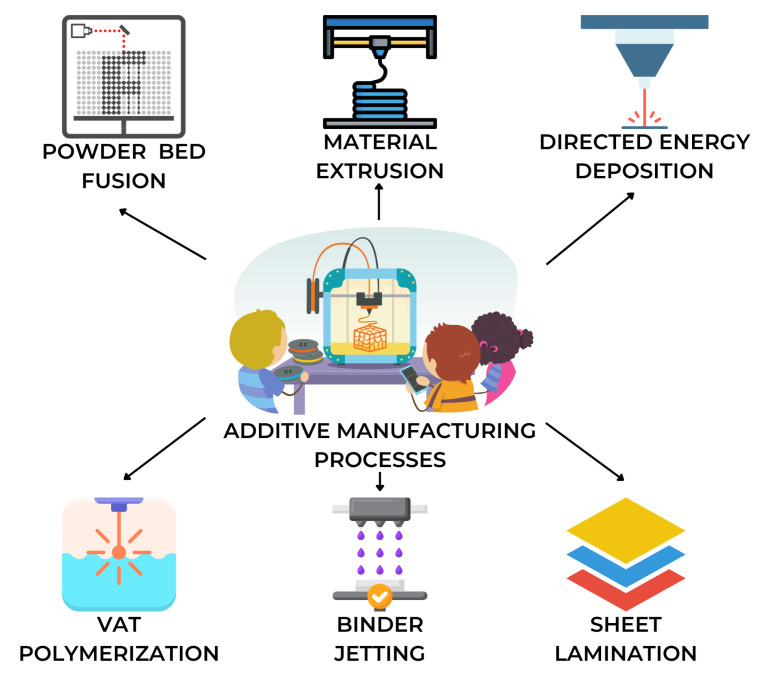
Additive manufacturing comes in many forms. Powder bed fusion, material extrusion, directed energy deposition, vat polymerization, binder jetting, and sheet lamination are the main processes in additive manufacturing.

**Figure 2 sensors-22-09446-f002:**
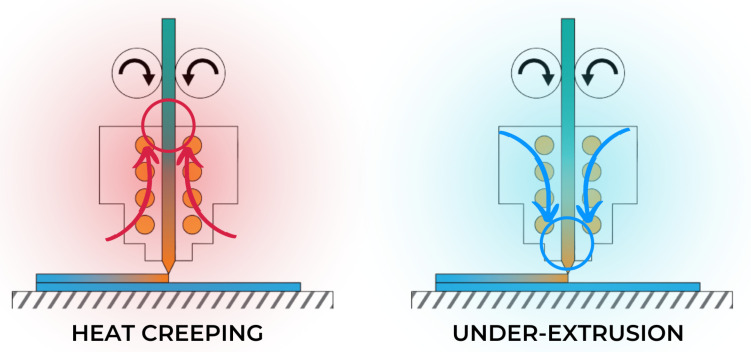
Overheating of plastic can cause a phenomenon called “heat creeping” where plastic tends to go back up due to the high temperature and causes clogging. Under-extrusion is the result of underheating, wherein the filament towards the nozzle solidifies and causes clogging.

**Figure 3 sensors-22-09446-f003:**
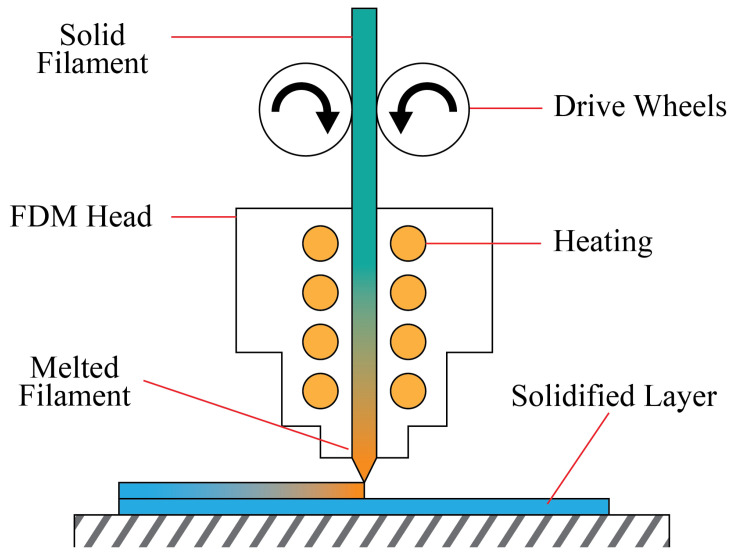
In printing using an FDM printer, drive wheels push a solid filament for the FDM head to melt and extrude the plastic.

**Figure 4 sensors-22-09446-f004:**
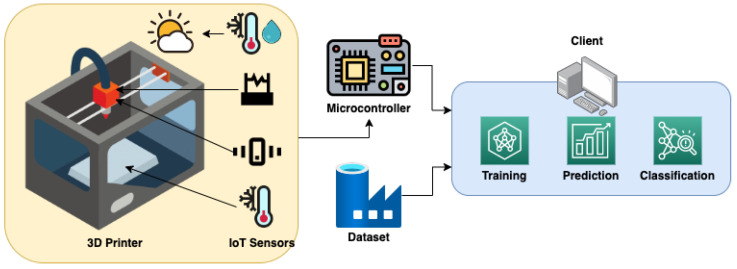
The FDM sensor-based fault monitoring framework involves the collection of data from collaborative IoT sensors and running them through a machine learning model.

**Figure 5 sensors-22-09446-f005:**
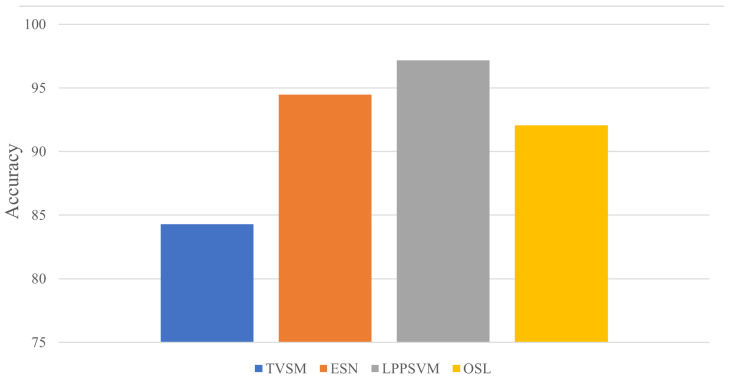
Comparison of novel sensor (magnetic field signal) [39] to conventional sensors (attitude sensor) in [41,44,46] for mechanical fault diagnosis of 3D delta printers.

**Table 1 sensors-22-09446-t001:** List of Data-Driven Solution Papers for FDM Printers.

Publication Year	FDM	Publications
2021	15	[2,3,28,29,30,31,32,33,34,35,36,37,38,39,40]
2020	4	[26,41,42,43]
2019	15	[10,22,24,44,45,46,47,48,49,50,51,52,53,54,55]
2018	12	[25,56,57,58,59,60,61,62,63,64,65,66]
2017	2	[67,68]
Total	48	

**Table 2 sensors-22-09446-t002:** Method Summary and Equipment Details of Mechanical Fault Diagnosis Studies.

PY	Ref	Printer Details	ST	Sensor Details	Material Used	Monitoring Type	Proposed Algorithm	Performance
2017	[70]	Hyrel3D E5 Engine FDM	AE	PAC 2/4/6, PAC PCI-2 MISTRAS differential	N/A	filament state	HSMM	Acc.: 91.9%
2018	[60]	Hyrel3D E5 Engine FDM	AE	AE sensor, PAC 2/4/6, PAC PCI-2	ABS	extruder state	CFSFDP	Block: 88% Semi-block: 79.53%
2018	[59]	Ultimater 2+ ASTM D 1708	Accel, AE	PCB Piezotronics 353B03 Physical	ABS	bolt state	SVM	87.5%
2019	[22]	delta 3D printer (SLD-BL600-6)	Attitude signal	BWT901 attitude sensor	N/A	joint bearing state	EFMSAE	N/A
2019	[23]	delta 3D printer	Attitude signal	BWT901 attitude sensor	N/A	joint bearing state	DFESN	Acc.: >90%
2019	[44]	delta 3D printer (SLD-BL600-6)	Attitude signal	BWT901 attitude sensor	N/A	joint bearing synchronous belt	ESN	97.17%
2019	[46]	delta 3D printer (SLD-BL600-6)	Attitude signal	BWT901 attitude sensor Endevco piezoelectric vibration sensor	N/A	joint bearing synchronous belt	TSVM	84.29%
2019	[45]	Markforged Two	Vibration signal	(7251A-500 1-channel & 65-10 3-channel)	Onyx	extruder state	BPNN, LS-SVM	normal: 95.56% warpage: 96% material stack: 100%
2020	[41]	delta 3D printer (SLD-BL600-6)	Attitude signal	BWT901 attitude sensor	N/A	joint bearing synchronous belt	LPPSVM	93.49%
2020	[26]	N/A	Vibration signal	N/A	N/A	roller fault	BCNN	Acc.: 99.82%
2021	[38]	Hyrel3D printer	AE	Mistagroup AE sensor PAC 2/4/6, PAC PCI-2	N/A	extruder state	Physics- constrained dictionary learning, K-SVD	normal (RMS): 4.6% material holding (RMS): 0% extruder blockade (RMS): 0% running out (RMS): 2.3%
2021	[39]	delta 3D printer (SLD-BL600-6)	Magnetic field	AK8963 electronic compass	N/A	joint bearing, synchronous belt	One-shot learning (OSL)	92.07%
2021	[35]	Prusa i3	Vibration signal	LSM330 accelerometer	N/A	extruder state	CNN	97.7%

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
