# Peer review of "Exploring Machine Learning-Based Fault Monitoring for Polymer-Based Additive Manufacturing: Challenges and Opportunities"

_sensors, 2022, doi:10.3390/s22239446_

Round 1

Reviewer 1 Report

The subject presented in this manuscript has great visibility and interest, and there are numerous review articles in the literature on this topic, published in the last two years. Review articles on “A review on machine learning in 3D printing: applications, potential, and challenges” and on “A review on machine learning in 3D printing” have been published recently.

This paper, although,  it has the advantage of taking an integrated approach to machine learning-based mechanical fault monitoring systems in fused deposition modeling. This interface currently has a redoubled interest in the new paradigm of Industry 4.0.

In this paper were collected relevant publications from 2017 to 2021.

In this manuscript, methodologies for integrating conventional methods ( such as  Sensor, Signal, and Data, Data-Driven Mechanical Fault Monitoring, , Acoustic Emission, Acceleration and Vibration, Magnetic Field, etc.) into Machine Learning algorithms are listed, highlighting their challenges and opportunities.

Only the Table 2 needs to be improvement. It seems to me that too much differentiation of font sizes, as it is e.g. in Table 1. 

Author Response

Thanks for your kind comments. Your letter indicated that our manuscript (sensors-2034728) should be revised in order to answer certain issues that have been addressed by the reviewers.   The paper was revised and improved according to the reviewers’ comments.
The followings are the revised contents and answers (italics) to each reviewers’ suggestions

Reviewer 2 Report

In the Introduction section, could you add some introduction about the other methods, especially the advantages and weakness when compared with ML-based methods, as you said in lines 40-41.

In the introduction section, please add a short discussion about the “Faults”.

Add some figures about the FDM process, monitoring techniques, will be helpful for readers to understand this manuscript. There are few figures in this manuscript.

For the predictive methods, it would be better if a comparison among different kinds of predictive methods can be provided. As we know, there are machine-learning methods, finite-element based numerical methods, and physics-based analytical methods in literature. All these methods can be used to predict the defects, and other phenomena in the additive manufacturing process. Could you please add a short discussion about the strengths and weaknesses of FEM and analytical methods, so that to show the meaning of developing machine learning methods? The following literature may be helpful:

https://doi.org/10.1016/j.matdes.2018.06.037

https://doi.org/10.1016/J.ENG.2017.05.023

https://doi.org/10.3390/app112412053

It will be better if the author can add a complete list of Nomenclature/abbreviations in this manuscript. The current list at the end of this manuscript seems not complete.

This manuscript focuses on the FDM process. Could you please add some discussion about the SLM process, and talk about whether and how the ML models for FDM can be extended to SLM process?

Author Response

Thanks for your kind comments. Your letter indicated that our manuscript (sensors-2034728) should be revised in order to answer certain issues that have been addressed by the reviewers.   The paper was revised and improved according to the reviewers’ comments.
The followings are the revised contents and answers (italics) to each reviewers’ suggestions.

Reviewer 3 Report

The review paper

“Exploring Machine Learning-Based Fault Monitoring for Polymer-Based Additive Manufacturing: Challenges and Opportunities”,

by Sampredo et al.,

reviews the recent contributions in the field of Machine Learning (ML)-based mechanical fault monitoring systems in fused deposition modelling (FDM), a specific technology for Additive Manufacturing (AM).

This review paper could be of great interest to researchers in the field of 3D printing and fault detection. Hence, it has very good potential.

However, the paper needs some improvements, both in its content and editing; these are enlisted here below.

1.      The query lines (and/or keywords) used for the research of the papers included in the document should be reported.

2.      Related to the previous remark, the inclusion/exclusion criteria considered for the selection of the 35 papers should be clearly reported.

3.      Table 1, it could be useful to add a third column with the references to the mentioned papers (divided accordingly to the year of publication).

4.      While graphically captivating, Figure 1 is a bit too generic and does not provide much helpful information in its current form.

5.      In the Introduction, it would be worth expanding the discussion about the potential applications of Fused Deposition Modeling (FDM) specifically and Additive Manufacturing (AM) in general. For instance, https://doi.org/10.3390/polym14132639 used AM to mimic the mechanical behaviour of human soft tissues. This and other mechanical and/or biomedical applications could be mentioned.

6.       Please double-check carefully the text for typos and grammar mistakes. For instance, in several cases, the blank space between a word and the following reference is missing (e.g. "sensors[30]”).

7.       “raspberry pi” and other commercial names should be capitalised (Raspberry Pi).

8.       It is a bit confusing that on pages 2 and 3 it is stated "relevant research papers are published between 2017 to 2021" and "our search did not produce any relevant results for 2017 publication”. The same statement is repeated in the Conclusions: “The researchers collected relevant publications from 2017 to 2021 and observed that 2017 returned nothing”.  Yet the Reference List includes [31] “Real-time FDM machine condition monitoring and diagnosis based on acoustic emission and hidden semi-Markov model” which was published in 2016. Please correct if needed.

If papers published before 2017 have been omitted for any specific reason, the text should clearly state this.

9.       Table 1 mentions a total of 35 documents but only 13 are enlisted in Table 2. Why?

10.   The paper would benefit from more Figures, to better portray the (general) AM and (specific) FDM processes.

11.   There is a bit of overlapping between Sec 5 Discussion and Sec 6 Conclusions. It would be better to move all the detailed discussions of the single papers to Section 5 and leave the Conclusions only for the most general and final comments.

Author Response

(The authors gave the same response as above.)

Round 2

Reviewer 2 Report

Thanks for the revisions. Please add a short discussion about physics-based analytical modeling method when you compare the machine-learning method with the FEM method. You mentioned that machine learning methods can give rapid solutions. The analytical modeling methods can also give rapid solutions, and the analytical methods do not rely on the plenty of experimental data. Then what is the advantage of machine learning methods when compared to analytical modeling methods? Could you add a short discussion about this? You should show clearly the meaning and unique strengths of machine learning methods, so that the readers can understand the meaning of your manuscirpt. The following literature are very helpful, even though they are about SLM, but the advantages and weakness of different methods are summarized well in these literature:

https://doi.org/10.1016/j.matdes.2018.06.037 - https://doi.org/10.1016/J.ENG.2017.05.023 - https://doi.org/10.3390/app112412053

Also, a little discussion about the material-wasting, time-consuming and expensive cost of experiments will help show the meaning of modeling work. 

Author Response

Thanks for your kind comments. Your letter indicated that our manuscript
(sensors-2034728) should be revised in order to answer certain issues that have been addressed by the reviewers. The paper was revised and improved according to the reviewers' comments.

Reviewer 3 Report

The authors have adequately faced all the observations made in the first round of reviews. Thus, this Reviewer suggests the acceptance of the submitted paper and its publication, after careful grammar checking and proofreading.

Author Response

We thank the reviewer for the kind and valuable suggestions.

Round 3

Reviewer 2 Report

I have read the answers from the authors. The manuscript can be published.